# Impact of Water Exposure and Temperature Changes on Skin Barrier Function

**DOI:** 10.3390/jcm11020298

**Published:** 2022-01-07

**Authors:** Manuel Herrero-Fernandez, Trinidad Montero-Vilchez, Pablo Diaz-Calvillo, Maria Romera-Vilchez, Agustin Buendia-Eisman, Salvador Arias-Santiago

**Affiliations:** 1Dermatology Department, Faculty of Medicine, University of Granada, 18012 Granada, Spain; manuherrero@correo.ugr.es (M.H.-F.); mariiarv@correo.ugr.es (M.R.-V.); abuendia@ugr.es (A.B.-E.); salvadorarias@ugr.es (S.A.-S.); 2Dermatology Department, Hospital Universitario Virgen de las Nieves, 18012 Granada, Spain; pdc.muro@gmail.com; 3Instituto de Investigación Biosanitaria Granada, 18012 Granada, Spain

**Keywords:** dermatological diseases, homeostasis, skin barrier, temperature, water

## Abstract

The frequency of hand hygiene has increased due to the COVID-19 pandemic, but there is little evidence regarding the impact of water exposure and temperature on skin. The aim of this study is to evaluate the effect of water exposure and temperature on skin barrier function in healthy individuals. A prospective observational study was conducted. Temperature, pH, transepidermal water loss (TEWL), erythema and stratum corneum hydration (SCH) were measured objectively before and after hot- and cold-water exposure and TempTest® (Microcaya TempTest, Bilbao, Spain) contact. Fifty healthy volunteers were enrolled. Hot-water exposure increased TEWL (25.75 vs. 58.58 g·h^−1^·m^−2^), pH (6.33 vs. 6.65) and erythema (249.45 vs. 286.34 AU). Cold-water immersion increased TEWL (25.75 vs. 34.96 g·h^−1^·m^−2^) and pH (6.33 vs. 6.62). TEWL (7.99 vs. 9.98 g·h^−1^·m^−2^) and erythema (209.07 vs. 227.79 AU) increased after being in contact with the hot region (44 °C) of the TempTest. No significant differences were found after contact with the cold region (4 °C) of the TempTest. In conclusion, long and continuous water exposure damages skin barrier function, with hot water being even more harmful. It would be advisable to use cold or lukewarm water for handwashing and avoid hot water. Knowing the proper temperature for hand washing might be an important measure to prevent flares in patients with previous inflammatory skin diseases on their hands.

## 1. Introduction

The skin is the largest organ in the human body and accomplishes regulatory and defensive functions that reside in the epidermis, mainly in the stratum corneum [1]. To evaluate the epidermal barrier, measurements of transepidermal water loss (TEWL) are usually performed [2]. A high TEWL is associated with a deficiency in the skin barrier [3]. Other parameters to assess epidermal function are the stratum corneum hydration (SCH), pH of the skin surface, temperature, erythema and melanin index [4]. These parameters can change depending on the anatomical location and can be affected by skin diseases such as psoriasis or atopic dermatitis [4]. Skin-related studies should not be based on a single parameter to cover all the functions of the epidermal barrier, as an integrated and multiparametric approach is needed to evaluate the properties of the skin barrier [5]. The epidermal barrier is necessary to maintain cutaneous homeostasis and protect the body against multiple external factors, such as environmental conditions, chemical stress and ultraviolet radiation [6].

Recently, the coronavirus 2019 (COVID-19) pandemic, caused by severe acute respiratory syndrome coronavirus 2 (SARS-CoV-2), has increased the frequency of hand hygiene in the overall population [7] as it is important to wash hands properly to avoid transmission of the virus [8]. However, frequent hand washing might also be challenging because hand hygiene products may damage skin [9]. Moreover, it has been noted that skin barrier dysfunction increases the expression of angiotensin-converting enzyme 2, the cell receptor for SARS-CoV-2, in the basal layer of the epidermis which may increase the risk of being infected with this virus [10,11,12].

Several risk factors are associated with skin damage, such as having a previous history of atopic dermatitis or hand eczema, female sex and longer working hours [13], but there is little research on the effect of water exposure and temperature [14,15]. Hot environments increase sweat production, which leads to higher hydration, TEWL, sebum production and greasiness, and lower pH [16]. It has been shown that water temperature does not have an impact on microbe removal [14,17] while the American Contact Dermatitis Society recommends washing hands with cold or lukewarm water to avoid skin irritation [18].

As there is no solid evidence regarding the effect of water exposure and temperature on skin, the objective of this study is to evaluate the impact of water exposure and temperature on cutaneous homeostasis and skin barrier function.

## 2. Materials and Methods

### 2.1. Design

A prospective observational study was conducted.

### 2.2. Study Population

Participants were recruited from February to May 2021 at the Dermatology Department of the Hospital Universitario Virgen de las Nieves in Granada.

Inclusion criteria: Healthy subjects, aged between 18 and 65 years who signed the informed consent form.Exclusion criteria: Subjects who did not sign the informed consent, subjects with an inflammatory skin disease (psoriasis, hidradenitis, atopic dermatitis) or another type of disease that may alter the epidermal barrier function and skin homeostasis; subjects receiving any topical, physical or systemic treatment which may alter the epidermal barrier function and skin homeostasis.

### 2.3. Study Variables Measurement

Sociodemographic variables (age and sex), smoking, skin care or use of body moisturizers were recorded by taking a clinical history. Biophysical parameters related to skin barrier function were measured by non-invasive tools. SCH was measured in Arbitrary Units (AU) using the Corneometer^®^ (CM 825, Bilbao, Spain); TEWL in g·h^−1^·m^−2^ with the Tewameter^®^ (TM 300, Bilbao, Spain); pH with the Skin-pH-Meter^®^ (PH 905, Bilbao, Spain); erythema index in AU using the Mexameter^®^ MX 18, and skin temperature in °C with the Skin-Thermometer ST 500; Bilbao, Spain. All these parameters were measured with the multi-probe adapter (MPA, Courage + Khazaka electronic GmbH, Bilbao, Spain). All measurements were taken in the same room at a mean room temperature of 23 ± 1 °C and ambient air humidity of 35% (range, 30–40%). All participants underwent an adaptation period of 20 min before the measurements were taken.

### 2.4. Water and Temperature Exposure

Skin barrier function parameters were measured at baseline on the right volar forearm and on both palms.

To evaluate the impact of water exposure on the epidermal barrier, participants immersed their right hand in hot water (41.29 (2.29 SD) °C) and their left hand in cold water (11.13 (2.71 SD) °C). They kept both hands submerged in 50 cm^3^ of water for 10 min. Skin barrier function parameters were measured again on both palms after this exposure (Figure 1).

To evaluate changes after contact with different temperatures, patients placed their forearm for five minutes on the TempTest^®^ (Microcaya TempTest, Bilbao, Spain) thermal element, which is used clinically to diagnose heat or cold contact urticaria [19,20]. The aluminium template of TempTest produces a temperature range continuously between 4 and 44 °C [21]. Skin barrier function parameters were measured at baseline and after the exposure on the regions of the volar forearm that were in contact with the aluminium template at 4 and at 44 °C (Figure 2).

The two exposures are not expected to interfere with each other. All participants immersed their hand in water first, whereas the volar forearm was not wet after water exposure, and there was an adaptation period of 20 min between the last measure on the palms and the baseline measure on the volar forearm.

### 2.5. Statistical Analysis

First, a descriptive analysis was carried out. Quantitative variables were expressed as means (standard deviation, SD) and qualitative variables as absolute (relative) frequency distributions. The normality of the variables was checked using the Shapiro–Wilk test. To compare continuous variables between independent samples, the Student’s *t*-test for independent samples or Welch’s test were used according to the homogeneity of the variances, previously evaluated by Levene’s test. The Student’s *t*-test for paired samples was used to compare variables in the same individuals. Statistical significance was considered when *p* < 0.05 with two tails. Statistical analyses were performed using SPSS version 24.0 (SPSS Inc., Chicago, IL, USA).

### 2.6. Ethics

All participants were informed of the study’s objectives when they signed the informed consent form to participate. This study was approved by the Ethics Committee of Hospital Universitario Virgen de las Nieves in Granada. All measurements were non-invasive, and the confidentiality of participant data was strictly preserved.

## 3. Results

### 3.1. Participants’ Characteristics

The study included 50 healthy individuals (20 men and 30 women) with a mean age of 33.3 (12.94 SD) years old. The demographic characteristics of the participants are displayed (Table 1).

### 3.2. Changes in Skin Barrier Function after Water Exposure

Differences in skin barrier parameters were found before and after water exposure (Figure 3, Appendix A). Temperature, pH, TEWL and SCH changed after cold water exposure. Temperature decreased (30.18 vs. 23.49 °C, *p* < 0.001) while pH (6.33 vs. 6.62, *p* < 0.001), TEWL (25.75 vs. 34.96 g·h^−1^·m^−2^, *p* < 0.001) and SCH (46.69 vs. 50.55 AU, *p* = 0.04) increased. No differences were found in erythema.

Temperature, pH, erythema and TEWL changed after hot water exposure. Temperature (30.18 vs. 32.60 °C, *p* < 0.001), pH (6.33 vs. 6.65, *p* < 0.001), TEWL (25.75 vs. 58.58 g·h^−1^·m^−2^, *p* < 0.001) and erythema (249.45 vs. 286.34 AU, *p* < 0.001) increased. No differences were found in SCH.

There were differences in temperature, erythema, TEWL and SCH between cold and hot water exposure. Temperature (23.49 vs. 32.60 °C, *p* < 0.001), TEWL (34.96 vs. 58.58 g·h^−1^·m^−2^, *p* < 0.001) and erythema values (253.63 vs. 286.34 AU, *p* < 0.001) were higher in hot water vs. cold water respectively, while SCH values (50.55 vs. 44.50 AU, *p* < 0.004) were lower. No differences were found in pH.

### 3.3. Changes in Skin Barrier Function after Direct Contact with Different Temperatures

Skin barrier function parameters also changed after direct contact with different temperatures (TempTest exposure) (Figure 4, Appendix A). Only temperature decreased after direct contact with cold temperature (exposure to TempTest at 4 °C) (30.76 vs. 25.35 °C, *p* < 0.001). No differences in pH, erythema, TEWL and SCH were found.

Temperature (30.76 vs. 32.45 °C, *p* < 0.001), TEWL (7.99 vs. 9.98 g·h^−1^·m^−2^, *p* < 0.001) and erythema (210.45 vs. 227.79 AU, *p* = 0.001) increased after direct contact with hot temperature (exposure to TempTest at 44 °C). No differences were found in pH or SCH.

There were differences between direct contact with cold and hot temperature. Temperature (25.35 vs. 32.45 °C, *p* < 0.001), TEWL (8.74 vs. 9.98 g·h^−1^·m^−2^, *p* = 0.003) and erythema values (209.07 vs. 227.79 AU, *p* = 0.017) were higher after contact with hot temperature vs. contact with cold temperature. No differences were found in pH or SCH.

### 3.4. Differences in Skin Barrier Changes between Sexes and Ages

Some differences were found in skin barrier changes between sexes (Table 2). The temperature decrease was higher in women than men (−7.29 vs−5.78, *p* = 0.02) after cold water. Erythema values on palm at baseline were higher in men (278.08 vs. 230.37 AU, *p* = 0.003) but changes were similar between sexes. SCH decreased in men while it increased in women after cold water (−1.96 vs. +7.74, *p* = 0.008). No differences were found in pH or TEWL.

Men showed higher temperature (31.20 vs. 30.47 °C, *p* = 0.038) and erythema (239.19 vs. 191.30 AU, *p* = 0.008) on the volar forearm at baseline but changes after TempTest were similar in both sexes. No differences in pH, TEWL or SCH were found.

Participants were classified according to their age: <30 years (56%, 28/50) or ≥30 years (44%, 22/50), as the turning points of skin barrier function appear in an individual’s thirties [22] (Table 3). Temperature (+ 2.60 vs. +2.20 °C, *p* = 0.012) and pH (+0.39 vs. +0.24, *p* = 0.04) increases were higher after hot water in participants < 30 years of age than in participants ≥30. Erythema (266.49 vs. 227.88 AU, *p* = 0.026) and SCH (53.62 vs. 37.87 AU, *p* = 0.002) on hands at baseline were higher in participants <30 than in participants ≥30, but changes after water exposure were similar between ages groups.

On the volar forearm, erythema was higher in participants <30 than in participants ≥ 30 (228.80 vs. 187.10, *p* = 0.015) but changes were similar between age groups after TempTest. No differences in other parameters were found.

## 4. Discussion

The skin barrier is impaired by water exposure, especially hot water. This is reflected in increased TEWL and pH values. Regarding temperature (TempTest exposure), only direct contact with hot temperatures showed skin barrier damage by increasing both TEWL and erythema. For the first time, we report skin barrier function changes after cold and hot water exposure and direct contact to cold and hot temperature.

Water exposure could damage the skin barrier through several mechanisms. Long exposure to water leads to disruption of the stratum corneum intercellular lipid lamellae, induces swelling in the corneocytes and the formation of large pools of water in the intercellular space [23,24]. Moreover, it has been described that water exposure causes changes in stratum corneum morphology and increases hydration in a dose-dependent way, facilitating the penetration of extrinsic irritants or allergens and providing a suitable environment for bacterial overgrowth [24,25]. A significant correlation has also been found between TEWL and temperature, as temperature promotes the mass transfer of water from the stratum corneum to the environment [26]. This could explain why TEWL values were higher after hot water exposure than after cold water exposure.

Water temperature might be a significant factor which increases skin damage when frequent handwashing is necessary, even more so nowadays because hand hygiene awareness has increased due to the COVID-19 pandemic [7]. Lipid fluidization, or disorganized lipid structure, is affected by higher water temperatures, resulting in greater skin permeability [17,27]. It is not recommended to use alcohol-based hand gels [28] or gloves with wet hands because it increases irritating substances trapped on the hands [18] that have been proven to disrupt the skin barrier [29]. Moreover, the American Contact Dermatitis Society recommends washing hands with cold or lukewarm water [18]. Our study supports all these recommendations with objective parameters for the first time, as we found that long and continuous water exposure could damage the skin barrier and that hot water is more aggressive than cold water, as shown by higher increases in TEWL values. It would be advisable to use cold or lukewarm water for handwashing and avoid hot water. Further research with different water temperature ranges should be conducted to select the most adequate temperature for handwashing.

Although this study did not include a pediatric population, the actual guidelines for skin care in infants recommends that the water temperature to wash children should be 37−37.5 °C [30]. This recommendation is based on expertise opinion, but we did not find any study comparing skin barrier function impairment with different water temperature in children. There might be differences in skin barrier function and water impact between children and adults as it has been described that pediatric skin undergoes a process of adaptation and maturation postnatally [31]. It could thus be interesting to develop similar studies in a pediatric population to evaluate the proper water temperature to wash children.

It has previously been shown that that TEWL and pH increased after immersing the volar forearm in tap water (15–20 °C) for 30 min a day for five consecutive days [24]. Nevertheless, the experimental conditions were different from ours and it is difficult to compare these studies. Other research has also showed that TEWL increased after water-patch occlusion [32], but this study is biased as occlusion alone can disrupt the skin barrier [33,34] (Table 4). For the first time, we report skin barrier function changes after water immersion and compare the effect of cold and hot water.

Regarding the direct effect of temperature on the skin, it has previously been observed that ambient air temperature affects skin properties such as pH, TEWL, sebum content, hydration, elasticity, wrinkles, skin pores and skin sensitivity [16,26,35]. This has been extensively studied in relation to the seasons of the year [16,26,35]. Sebum output, melanin content of pigmented spots, skin colour and skin hydration are increased in summer compared to in winter [35]. Higher ambient air temperature also positively correlates to high TEWL values [16,26] and promotes sweating, which increases hydration and sebum secretion and decreases pH [16] (Table 4). We found that temperature, TEWL and erythema increase after exposure to heat. We did not find any differences in pH or SCH, which may be explained by a short contact time or by differences in the type of temperature exposure. Our research assessed the impact on skin barrier function after direct contact with different temperatures while the others evaluated changes after ambient air. We did not find any study that compared epidermal barrier changes after direct contact with different temperatures. It would be interesting to develop further research to compare skin barrier impairment between ambient air exposure and direct contact with different temperatures.

As far as we know, no previous study included information regarding changes after water and temperature exposure between sexes and ages. We observed that women were more prone to have changes in temperature and SCH after cold water exposure, being likely explained by the slightly thicker stratum corneum in males [36]. However, the relationship between gender and skin barrier function seems to be outweighed by individualized factors [36]. In relation to different impact of water and temperature depending on age, we only found that temperature and pH increases were higher in the younger group after water exposure. One explanation could be that young individuals have a greater blood flow, especially in sites exposed to the environment like the hands, what would rapidly adapt the skin to the environmental conditions [37].

Chronic skin diseases which typically appear on the hands are contact dermatitis (both irritative and allergic), atopic dermatitis and palmoplantar psoriasis [38,39,40]. Long water exposure in these patients is related to disease worsening [41], as immersing irritated hands in water increases blood flow and aggravates inflammation [42]. Moreover, it has been observed that high water temperature also impacts on the irritant capacity of detergents and could increase the risk for developing irritant dermatitis [43]. Therefore, washing hands at a suitable temperature might be a simple but essential step for preventing disease flares in patients with hand skin diseases.

Our study shows that TEWL values in healthy individuals differ widely between the volar forearm and the palms, as has been reported previously [2,44]. This fact has been attributed to differences in the number of corneocyte cell layers and in the size and turnover rates of corneocytes [45], as well as the low amount of barrier lipids in the stratum corneum in the palm [46]. Moreover, the higher TEWL values on the palms could be due to their thicker stratum corneum, higher exposure to friction and damage, and greater density of eccrine sweat glands [47].

This study was subject to some limitations such as a limited sample size, the lack of a longer follow-up and the exposure time. It has been noted that a healthcare worker could wash their hands on average from 5 to as many as 42 times [7,48]. Moreover, WHO guidelines recommend that the whole procedure of hand washing should take 40–60 s [48]. Considering the lowest rates of hand washing, a person could wash his hands 10 times for 40 s each, exposing his hands to water around 8 min per day on average. It should be also considered that times of hand washing could impact on skin barrier function. This fact is not expected to change our conclusions because in our study we compared participants before and after each exposure, using paired samples, but it should be considered when designing this type of research. Future research could measure more parameters of skin barrier function and lengthen exposure time to both water and temperature as longer exposure would lead to even greater changes in skin barrier function parameters.

## 5. Conclusions

Water and temperature have a significant effect on the skin barrier. Skin is impaired by water exposure, even more so with hot water, as reflected by increased TEWL values. Hot temperature also damages the skin, as shown by increased erythema and TEWL values. Knowing the proper temperature for hand washing might be an important measure to prevent flares in patients with previous skin inflammatory diseases on their hands.

## Figures and Tables

**Figure 1 jcm-11-00298-f001:**
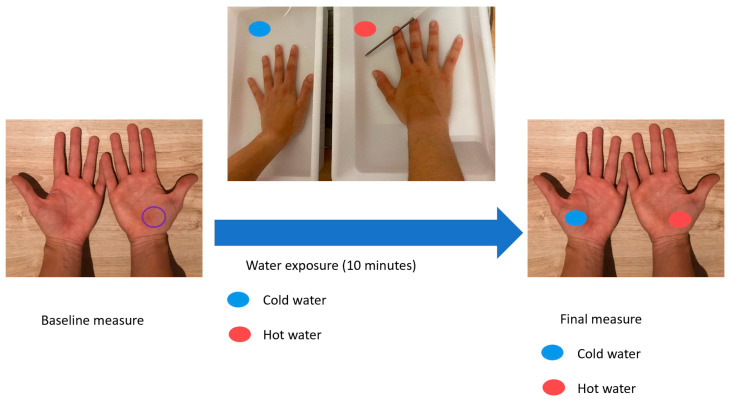
Flow chart of the measurements taken before and after water exposure.

**Figure 2 jcm-11-00298-f002:**
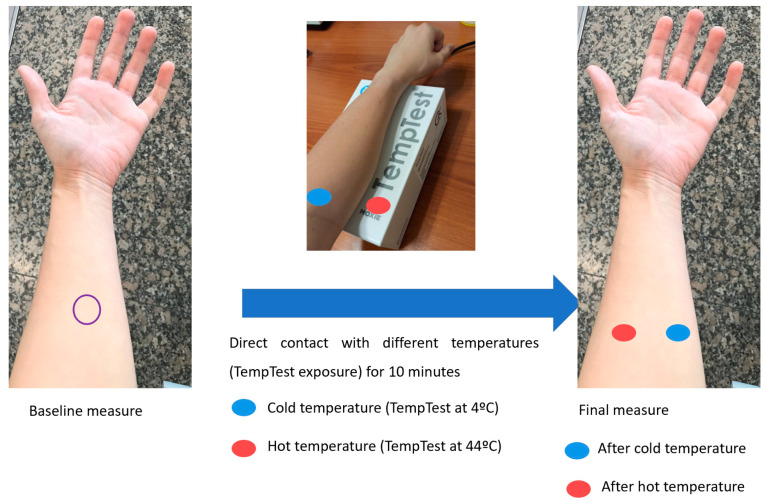
Flow chart of the measurements taken before and after direct contact with different temperatures (TempTest exposure).

**Figure 3 jcm-11-00298-f003:**
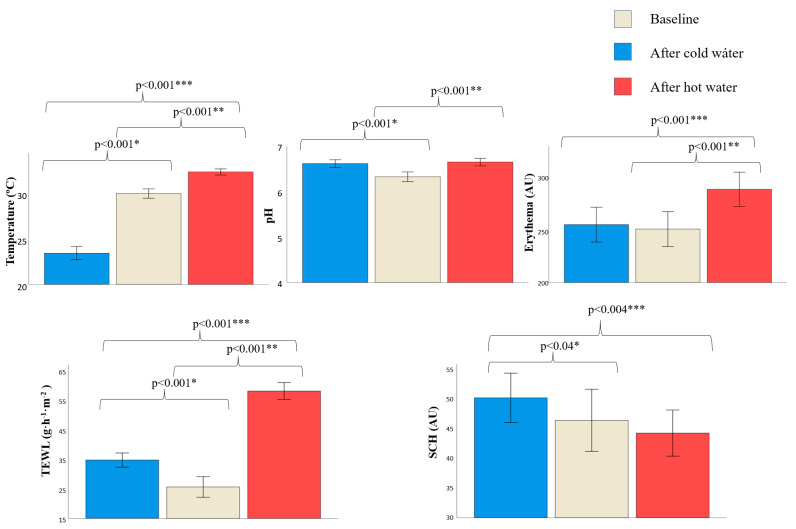
Skin barrier function parameters on the hands at baseline and after hot and cold-water exposure. Data is expressed as mean (standard deviations). AU, arbitrary units; SCH, stratum corneum hydration; TEWL, transepidermal water loss. * *p*-value after using Student’s *t*-test for paired samples to compare skin barrier function between baseline and after cold water exposure. ** *p*-value after using Student’s *t*-test for paired samples to compare skin barrier function between baseline and after hot water exposure. *** *p*-value after using Student’s *t*-test for paired samples to compare skin barrier function after cold and hot water exposure. Only significative *p*-values are shown (*p* < 0.05).

**Figure 4 jcm-11-00298-f004:**
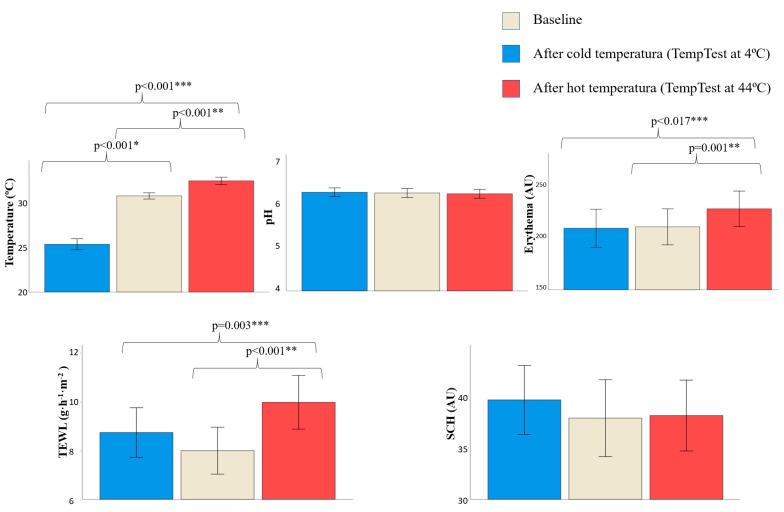
Skin barrier function parameters on the volar forearm at baseline and after direct contact to different temperatures. Data is expressed as mean (standard deviations). AU, arbitrary units; SCH, stratum corneum hydration; TEWL, transepidermal water loss. * *p*-value after using Student’s *t*-test for paired samples to compare skin barrier function between baseline and contact with cold temperatura (TempTest at 4 °C). ** *p*-value after using Student’s *t*-test for paired samples to compare skin barrier function between baseline and after contact with hot temperature (TempTest at 44 °C). *** *p*-value after using Student’s *t*-test for paired samples to compare skin barrier function after contact with cold and hot temperature. Only significative *p*-values are shown (*p* < 0.05).

**Table 1 jcm-11-00298-t001:** Descriptive characteristics of the sample.

Sociodemographic Features	Study Population (N = 50)
Age (years)	33.3 (12.94)
Gender	
Male	20 (40%)
Female	30 (60%)
Phototype	
I	1 (2%)
II	14 (28%)
III	29 (58%)
IV	5 (10%)
VI	1 (2%)
Smoking habit (yes)	4 (8%)
Cigarettes per day	7.25 (3.78)
Residence	
City	45 (90%)
Country	5 (10%)
Regular moisturizing	34 (68%)
Regular sunlight exposure	4 (8%)
Sun protection	
Always	21 (42%)
Sometimes	20 (40%)
Never	9 (18%)
Occupational category	
Doctors	23 (46%)
Nurses	18 (36%)
Miscellaneous	9 (18%)

Data is expressed as relative (absolute) frequencies and means (standard deviations).

**Table 2 jcm-11-00298-t002:** Changes in skin barrier function after hot and cold and water exposure, stratified by sex.

	Men (*n* = 20)			Women (*n* = 30)					
Water Exposure									
	Basal	Change after Cold Water	Change after Hot Water	Basal	Change after Cold Water	Change after Hot Water	*p* ^B^	*p* ^C^	*p* ^H^
**Temperature (°C)**	30.39 (2.04)	−5.78 (0.51)	+2.31 (0.30)	+30.03 (1.71)	−7.29 (0.37)	+2.50 (0.26)	0.523	0.020	0.647
**pH**	6.30 (0.43)	+0.37 (0.37)	+0.41 (0.10)	+6.35 (0.34)	+0.23 (0.06)	0.26 (0.06)	0.645	0.165	0.215
**Erythema (AU)**	278.08 (46.99)	+10.27 (10.75)	+31.14 (10.42)	+230.37 (56.12)	+0.12 (11.31)	+40.71 (9.82)	0.003	0.539	0.519
**TEWL** **(g·h^−1^·m^−2^)**	24.35 (10.88)	+12.21 (2.45)	+33.30 (2.36)	26.68 (13.41)	+7.21 (2.30)	+32.51 (2.32)	0.521	0.154	0.814
**SCH (AU)**	49.54 (16.51)	−1.96 (2.51)	−3.21 (2.69)	44.80 (20.13)	+7.74 (2.31)	−1.50 (3.22)	0.386	0.008	0.685
**TempTest Exposure**									
	**Basal**	**Change after 4 °C TempTest**	**Change after 44 °C TempTest**	**Basal**	**Change after 4 °C TempTest**	**Change after 44 °C TempTest**	** *p* ^Bt^ **	** *p* ^Ct^ **	** *p* ^Ht^ **
**Temperature (°C)**	31.20 (1.24)	−5.96 (0.56)	+1.58 (0.28)	30.47 (1.16)	−5.06 (0.34)	+1.76 (0.23)	0.038	0.178	0.624
**pH**	6.25 (0.39)	+0.04 (0.03)	−0.03 (0.03)	6.28 (0.37)	−0.01 (0.03)	−0.02 (0.03)	0.787	0.275	0.830
**Erythema (AU)**	239.19 (63.79)	+7.68 (12.94)	+13.41 (7.16)	191.30 (51.39)	−7.41 (8.14)	+19.95 (6.51)	0.008	0.303	0.512
**TEWL** **(g·h^−1^·m^−2^)**	7.01 (3.29)	+1.79 (0.87)	+2.74 (0.07)	8.65 (3.34)	+0.05 (0.58)	+1.48 (0.67)	0.092	0.089	0.216
**SCH (AU)**	34.21 (10.67)	−1.69 (1.65)	+1.99 (1.47)	40.14 (13.91)	+1.58 (1.30)	+1.53 (1.79)	0.095	0.841	0.216

Data is expressed as mean (standard deviation). AU, arbitrary units; SCH, stratum corneum hydration; TEWL, transepidermal water loss. Data is expressed as means (standard deviations, SD). ^B^
*p*-value after using Student’s *t*-test for independent samples to compare skin barrier function at baseline between men and women. ^C^
*p*-value after using Student’s *t*-test for independent samples to compare skin barrier function changes after cold water exposure between men and women. ^H^
*p*-value after using Student’s *t*-test for independent samples to compare skin barrier function changes after hot water exposure between men and women. ^Bt^
*p*-value after using Student’s *t*-test for independent samples to compare skin barrier function at baseline between men and women. ^Ct^
*p*-value after using Student’s *t*-test for independent samples to compare skin barrier function changes after 4 °C TempTest exposure between men and women. ^Ht^
*p*-value after using Student’s *t*-test for independent samples to compare skin barrier function changes after 44 °C TempTest exposure between men and women.

**Table 3 jcm-11-00298-t003:** Changes in skin barrier function after hot and cold water exposure, stratified by age.

	Age < 30 (*n* = 28)			Age ≥ 30 (*n* = 22)					
Water Exposure									
	Basal	Cold Water	Hot Water	Basal	Cold Water	Hot Water	*p* ^B^	*p* ^C^	*p* ^H^
**Temperature (** **°C)**	30.02 (1.80)	−5.99 (0.41)	+2.60 (0.26)	30.37 (1.90)	−7.57 (0.44)	+2.20 (0.30)	0.507	0.322	0.012
**pH**	6.29 (0.43)	+0.38 (0.07)	+0.39 (0.07)	6.37 (0.30)	+0.18 (0.06)	+0.24 (0.08)	0.406	0.153	0.040
**Erythema (AU)**	266.49 (40.45)	+2.66 (8.33)	+35.18 (8.74)	227.88 (68.29)	+6.11 (14.99)	+39.05 (12.16)	0.026	0.792	0.833
**TEWL (g·h^−1^·m^−2^)**	25.36 (12.14)	+10.79 (2.06)	+33.75 (2.06)	26.25 (12.99)	+7.20 (2.02)	+31.65 (2.63)	0.804	0.528	0.302
**SCH (AU)**	53.62 (18.46)	+1.75 (2.52)	−5.91 (3.09)	37.87 (15.33)	+6.55 (2.61)	+2.56 (2.85)	0.002	0.055	0.196
**TempTest exposure**									
	**Basal**	**4 °C TempTest**	**44 °C TempTest**	**Basal**	**4 °C TempTest**	**44 °C TempTest**	** *p* ^Bt^ **	** *p* ^Ct^ **	** *p* ^Ht^ **
**Temperature (°C)**	30.89 (1.12)	−5.68 (0.48)	+1.75 (0.25)	30.59 (1.38)	−5.21 (0.39)	+1.64 (0.25)	0.399	0.778	0.446
**pH**	6.25 (0.39)	−0.03 (0.03)	−0.02 (0.03)	6.28 (0.36)	−0.05 (0.03)	−0.02 (0.04)	0.803	0.936	0.065
**Erythema (AU)**	228.80 (59.38)	−5.10 (9.42)	+19.43 (7.66)	187.10 (55.56)	+1.54 (10.45)	+15.69 (6.26)	0.015	0.705	0.648
**TEWL (g·h^−1^·m^−2^)**	7.21 (2.39)	−0.08 (0.87)	+2.50 (0.82)	8.99 (4.19)	+1.40 (0.57)	+1.57 (0.62)	0.087	0.359	0.163
**SCH (AU)**	34.76 (10.53)	+1.20 (1.68)	+0.97 (2.04)	41.59 (14.85)	+2.18 (1.13)	−0.32 (1.63)	0.063	0.618	0.621

Data is expressed as mean (standard deviations). AU, arbitrary units; SCH, stratum corneum hydration; TEWL, transepidermal water loss. Data is expressed as means (standard deviations, SD). ^B^
*p*-value after using Student’s *t*-test for independent samples to compare skin barrier function at baseline between <30 and ≥30 years old. ^C^
*p*-value after using Student’s *t*-test for independent samples to compare skin barrier function after cold water exposure between <30 and ≥30 years old. ^H^
*p*-value after using Student’s *t*-test for independent samples to compare skin barrier function after hot water exposure between <30 and ≥30 years old. ^Bt^
*p*-value after using Student’s *t*-test for independent samples to compare skin barrier function at baseline between <30 and ≥30 years old. ^Ct^
*p*-value after using Student’s *t*-test for independent samples to compare skin barrier function after 4 °C TempTest exposure between <30 and ≥30 years old. ^Ht^
*p*-value after using Student’s *t*-test for independent samples to compare skin barrier function after 44 °C TempTest exposure between <30 and ≥30 years old.

**Table 4 jcm-11-00298-t004:** Studies evaluating water and temperature exposure.

Study	Number of Participants	Exposure	Results
Water Exposure			
Firooz et al. 2015 [24]	20	Water immersion at 15–20 °C for 30 min a day for five days	TEWL increased pH increased
Agner et al. 1993 [32]	14	Water patch-occlusion during 24 h: - One closed patch test with 60 uL of an aqueous solution of 0.5% SLS on a filter disc. - One closed patch test with 60 uL sterile water on a filter disc. - One empty chamber	TEWL increased immediately and 30 min after removal of all test chambers. TEWL also increased 180 min after SLS and water patch removal.
Our study	50	Cold water exposure (11.13 (2.71 SD) °C) for 10 min	Temperature decreased by 6.69 °C pH increased by 0.29 TEWL increased by 9.21 (g·h^−1^·m^−2^) SCH increased 3.86 AU
		Hot water exposure (41.29 (2.29 SD) °C) for 10 min	Temperature increased by 2.42 °C pH increased by by 0.32 Erythema increased by 36.88 AU TEWL increased by 32.83 (g·h^−1^·m^−2^)
**Temperature exposure**			
Kim et al. 2019 [16]	20	Exposure to outdoor environment for 90 min (34.76 ± 2.79 °C, 53.13 ± 8.78% RH)	SCH on the forearm and sebum secretion on the face increased. SCH and TEWL on the cheek and greasiness on the forearm and the forehead decreased. pH decreased in the face and the forearm.
		Exposure to indoor environment for 90 min (22.97 ± 0.74 °C, 53.14 ± 2.37% RH)	SCH in the forehead and the forearm. Sebum secretion and greasiness on the face increased. TEWL and pH in the face and the forearm decreased.
Cravello et al. 2008 [26]	6	Exposure to three levels of ambient temperature (20 °C, 25 °C and 30 °C) and four levels of RH (25%, 45%, 65% and 85%)	TEWL is positively correlated to ambient temperature. Skin temperature is correlated positively to ambient temperature but not to RH. SCH is strongly affected by ambient temperature and RH.
Qiu et al. 2011 [35]	354	6 month periods, summer (35–40 °C, RH ≥ 70%) and winter (0–5 °C, RH ≥ 70%)	SCH and melanin increased in summer compared to winter.
Our study	50	Cold temperatrue exposure (4 °C)	Temperature decreased by 5.41 °C
		Hot temperature exposure (44 °C)	Temperatrure increased by 1.69 °C Erythema increased by 17.34 AU TEWL increased by 1.98 (g·h^−1^·m^−2^)

AU, arbitrary units; RH, relative humidity; SCH, stratum corneum hydration; SLS, sodium lauryl sulphate; TEWL, transepidermal water loss.

## Data Availability

The data presented in this study are available on request from the corresponding author.

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
