# Peer review of "Impact of Water Exposure and Temperature Changes on Skin Barrier Function"

_jcm, 2022, doi:10.3390/jcm11020298_

Round 1
Reviewer 1 Report
Dear Authors, Thank you for the opportunity to review your work. Please see my comments below. 1.The authors reference their study in relation to frequent hand washing in the COVID area yet they expose skin to water for 10 minutes. Hand washing only exposes the skin to water for 60-90 seconds (or shorter in most cases), much less than studied in this work. Even if talking about repeated hand washings throughout the day, the experimental setting is very far from real life exposure. Please address this issue in details.2.I am struggling to understand the exact interventions and measurements performed on the hands and volar forearms and I am afraid other readers may have a similar problem when reading this manuscript. Please create a flowchart that shows clearly in a succinct way what was done to what skin surface for how long and what was measured. 3.Also, there are several studies measuring temperature, pH, TEWL in similar settings, please make it very clear by a side--by-side comparison how this study is different from others. May even make a small table to summarize already existing literature on this topic. 4.The manuscript needs extensive English editing, there are several grammatical errors throughout.
Author Response
Reviewer: 1
Dear Authors, Thank you for the opportunity to review your work. Please see my comments below.
Thank you for your comments
1.The authors reference their study in relation to frequent hand washing in the COVID area yet they expose skin to water for 10 minutes. Hand washing only exposes the skin to water for 60-90 seconds (or shorter in most cases), much less than studied in this work. Even if talking about repeated hand washings throughout the day, the experimental setting is very far from real life exposure. Please address this issue in details.
Following your recommendation this issue has been included as a limitation of our study. Our aim was to analyse the impact of cold and hot water on skin barrier function and the effect of direct contact to different temperatures independently of the effect of water (TempTest exposure). It is true that one hand-washing is not going to last 10 minutes but a healthcare worker could wash their hands a on average from 5 to as many as 42 times as has been described in observational studies conducted in hospitals (10.1684/ejd.2020.3923 and WHO recommendations for hand washing available on https://www.who.int/publications/i/item/9789241597906). Moreover, WHO guidelines recommend that the whole procedure of hand washing should take 40-60 seconds. Considering the lowest rates of hand washing, a person could wash his hand 10 times for 40 seconds, exposing his hands to water around 8 minutes per day on average.
2.I am struggling to understand the exact interventions and measurements performed on the hands and volar forearms and I am afraid other readers may have a similar problem when reading this manuscript. Please create a flowchart that shows clearly in a succinct way what was done to what skin surface for how long and what was measured.
We have changed the figure 1 to the flow chart recommended. We trust that the figure has increased the comprehension of our study methods.
3.Also, there are several studies measuring temperature, pH, TEWL in similar settings, please make it very clear by a side--by-side comparison how this study is different from others. May even make a small table to summarize already existing literature on this topic.
Following your recommendations, we have included a table with the studies that evaluated skin barrier function changes after temperature an water exposure.
Table 3. Studies evaluating water and temperature exposure.
|
Study |
Number of participants |
Exposure |
Results |
|
Water exposure |
|||
|
Firooz et al. 2015 [24] |
20 |
Water immersion at 15-20ºC for 30 min a day for five days |
TEWL increased pH increased |
|
Agner et al. 1993 [32] |
14 |
Water patch-occlusion during 24 hours: - One closed patch test with 60uL of an aqueous solution of 0.5% SLS on a filter disc. - One closed patch test with 60uL sterile water on a filter disc. - One empty chamber |
TEWL increased immediately and 30 minutes after removal all test chambers. TEWL also increased 180 minutes after SLS and water patches removal.
|
|
Our study |
50 |
Cold water exposure (11.13 (2.71 SD) ºC) for 10 minutes |
Temperature decreased by 6.69ºC pH increased by 0.29 TEWL increased by 9.21 (g·h-1·m-2) SCH increased 3.86 AU |
|
Hot water exposure (41.29 (2.29 SD) ºC) for 10 minutes |
Temperature increased by 2.42ºC pH increased by by 0.32 Erythema increased by 36.88 AU TEWL increased by 32.83 (g·h-1·m-2) |
||
|
Temperature exposure |
|||
|
Kim et al. 2019 [16] |
20 |
Exposure to outdoor environment for 90 min (34.76±2.79°C, 53.13±8.78% RH) |
SCH on the forearm and sebum secretion on the face increased. SCH and TEWL on the cheek and grassiness on the forearm and the forehead decreased. pH decreased in the face and the forearm.
|
|
Exposure to indoor environment for 90 min (22.97±0.74°C, 53.14±2.37% RH) |
SCH in the forehead and the forearm. Sebum secretion and greasiness on the face increased. TEWL and pH in the face and the forearm decreased. |
||
|
Cravello et al. 2008 [26] |
6 |
Exposure to three levels of ambient temperature (20ºC, 25ºC and 30ºC) and four levels of RH (25%, 45%, 65% and 85%) |
TEWL is positively correlated to ambient temperature. Skin temperature is correlated positively to ambient temperature but not to RH. SCH is strongly affected by ambient temperature and RH. |
|
Qiu et al. 2011 [35] |
354 |
6 month periods, summer (35–40ºC, RH≥70%) and winter (0–5ºC, RH≥70%) |
SCH and melanin increased in summer compared to winter. |
|
Our study |
50 |
Cold temperatrue exposure (4ºC) |
Temperature decreased by 5.41ºC |
|
Hot temperature exposure (44ºC) |
Temperatrure increased by 1.69ºC Erythema increased by 17.34 AU TEWL increased by 1.98 (g·h-1·m-2) |
||
AU, arbitrary units; RH, relative humidity; SCH, stratum corneum hydration; SLS, sodium lauryl sulphate; TEWL, transepidermal water loss.
4.The manuscript needs extensive English editing, there are several grammatical errors throughout.
An English native speaker, expertise in scientific literature, has checked the language of the manuscript

Reviewer 2 Report
Nothing new. Hot water bath should be avoided in infants and persons with interrupted skin barrier like AD patients. It is well known and described in guidelines e.g. Water temperature should be 37°C–37.5°C in Pediatr. Derm. 2016 May;33(3):311-21. Recommendations from a European Roundtable Meeting on Best Practice Healthy Infant Skin Care
Author Response
Nothing new. Hot water bath should be avoided in infants and persons with interrupted skin barrier like AD patients. It is well known and described in guidelines e.g. Water temperature should be 37°C–37.5°C in Pediatr. Derm. 2016 May;33(3):311-21. Recommendations from a European Roundtable Meeting on Best Practice Healthy Infant Skin Care
Thank you for your comments. In this guideline it is recommended that the water temperature to wash children should be 37ºC-37.5ºC. This recommendation is based on expertise advice, with a lower level of evidence than prospective studies or cross-sectional studies. We have reviewed the literature of this guideline and that of Blume-Peytavi et al. 2009 and I have not found any study assessing objectives changes in skin barrier function after cold and hot water immersion. We found several articles evaluating TEWL after different type of cleansing procedures and skin care regimens to bathing (10.1159/000235552, 10.1111/1552-6909.12015, 10.1111/j.1525-1470.2009.01068.x, 10.1007/s13312-010-0161-8), and other assessing axillary temperature in newborn after bath (10.1016/s0897-1897(95)80591-5, 10.1111/j.1552-6909.2000.tb02753.x). We observed in an adult population that hot water could be more harmful than cold water. It has been described that pediatric skin undergoes a process of adaptation and maturation postnatally (10.1046/j.1525-1470.2002.00082.x), so there might be differences in skin barrier function and water impact between children and adults. It could be interesting to develop similar studies in a pediatric population to evaluate the proper water temperature to wash children. Following your recommendation, we have included this information in the discussion and cited the Recommendations from a European Roundtable Meeting on Best Practice Healthy Infant Skin Care. The following sentences have been added: “Although this study did not include a pediatric population, the actual guidelines for skin care in infants recommends that the water temperature to wash children should be 37ºC-37.5ºC[29]. This recommendation is based on expertise opinion, but we did not find any study comparing skin barrier function impairment with different water temperature in children. There might be differences in skin barrier function and water impact between children and adults as it has been described that pediatric skin undergoes a process of adaptation and maturation postnatally[30]. So, it could be interesting to develop similar studies in a pediatric population to evaluate the proper water temperature to wash children”.
Reviewer 3 Report
Please see attachment.

Author Response
Reviewer 3
The paper explained the extreme water temperatures may do harmful for skin barrier condition. This is an interesting idea, however there are quite many things to be addressed so proper conclusion can be drawn.
Thank you very much for all the comments.
Comments.
- We don’t usually write p value and data on abstract.
We have omitted p values in the abstract.
- Sociodemografic variables: how about job? I think this is an important factor that surely affects the skin barrier condition
It is true that the job could affects skin barrier condition. Participants were healthcare workers from our hospital, so the results are more homogeneous because different professions are not being compared. A total of 46% (23/50) were doctors, 36% (18/50) were nurses, and 18% (9/50) were miscellaneous healthcare workers, including nurses' assistant and administrative staff. This information has been added in table 1. Thank you for the comment.
- How did you evaluate the differences in temperature, erythema, TEWL and SCH between cold and hot water exposure (line 139 140; 153 155)?
To evaluate the differences in temperature, erythema, TEWL and SCH between cold and hot water exposure we used Student’s t-test for paired samples to compare skin barrier function after cold and after hot water exposure. This information is in the figure and we have also added it as a footnote in word format now.
Please explain what do you refer with evaluating changes after contact to different temperature by using TempTest (line 94 95). Line 98: 4–44°C do you refer to 4 and 44 °C
We evaluated temperature changes after contact with TempTest on the regions on the volar forearm that were in contact with the aluminium template at 4 and at 44 °C. We have modified this paragraph trying to make it clearer: “To evaluate changes after contact with different temperatures, patients placed their forearm for five minutes on the TempTest® (Microcaya TempTest, Bilbao, Spain) thermal element, which is used clinically to diagnose heat or cold contact urticaria[19, 20]. The aluminium template of TempTest produces a temperature range continuously between 4 and 44 °C [21], Figure 1B. Skin barrier function parameters were measured at baseline and after the exposure on the regions of the volar forearm that were in contact with the aluminium template at 4 and at 44 °C”.
- Table 1: the pB data of erythema had significant difference between men and women after cold and hot water . Also, the pBt of temperature and erythema showed significant differences. I think it is not appropriate to compare treatments on 2 subjects which already had differences at steady state. A same question goes to t able 2: pB of erythema and SCH, and pBt of erythema
You are right that it is not appropriate to compare the effects on 2 subjects with differences at steady state. To provide more suitable data, we have now compared the change (skin homeostasis parameters after the exposure – baseline) between sexes and between ages. We have also changed columns and have provided the change after each exposure instead of the final value trying to make it more understandable. we have also modified the corresponding paragraphs
- Why did you determine 30 years old as cut off of your study’s group?
We decided to determine 30 years old as cut off of our study group because the turning points of skin barrier function appears in an individual’s thirties (Pan et al. 2020 doi: 10.2147/CCID.S286402). We have included this information in the manuscript.
- In the first paragraph of discussion, you said that only the contact with hot temperature significantly damage the skin barrier. However, data of fig.2 showed significant differences between baseline and cold water exposure. So I would assume that cold temperature also induces skin barrier damage. What do you think about this?
In this paragraph we wanted to make difference between water exposure and direct contact with the aluminium template of TempTest. The first two sentences refer to changes after water exposure (new fig 3) and the last one to TempTest exposure (new fig 4). We only observed a decreased in temperature after direct contact with the aluminum template at 4ºC without differences in other skin barrier function parameters. We have changed these sentences trying to make them more understandable: “The skin barrier is impaired by water exposure, especially hot water. This is reflected in increased TEWL and pH values. Regarding temperature (TempTest exposure), only direct contact with hot temperatures showed skin barrier damage by in-creasing both TEWL and erythema”.
- The “employ” word in line 218: what does it mean?
This was a typographical mistake. We have deleted it. Moreover, Charlotte Bower, an English native speaker, expertise in scientific literature, has checked the language of the manuscript.
- In the discussion line 227: The experiment condition in ref.23 and yours are different. So I don’t think you can compare those results.
We have changed this paragraph according to your recommendations: “Previously, it has been shown that that TEWL and pH increased after immersing the volar forearm in tap water (15–20 °C) for 30 min a day for five consecutive days [23]. Nevertheless, the experiment conditions were different from ours and it is difficult to compare both studies”.
- You used the “what” word several times in the discussion, such as line 236. I think it is not suitable to use it on those sentences.
We have omitted “what” in several sentences and an English native speaker has also checked the manuscript.
- Discussion, line 240–247: If the differences between ages and sexes are already known, so what do you think a new findings of your study?
There are studies that compare skin barrier function between sex and ages but we did not find any that compare the different impact of water and temperature between sex and ages. We have included the following sentences: “As far as we know, no previous study included information regarding changes after water and temperature exposure between sexes and ages. We observed that women were more prone to have changes in temperature and SCH after cold water exposure, being likely explained by the slightly thicker stratum corneum in males [40]”.
Discussion, line 256–257: It is ok to have similar results with other studies, however it is better to also disclose unknown facts. I suggest that you show new findings of your data which has not been kn own previously from other papers.
We did not find any research that compare skin barrier function changes between cold and hot water exposure. Moreover, we did not find any article that compare skin barrier function changes between direct contact to cold and hot temperature. We have included this information in the first paragraph of the discussion: “For the first time we report skin barrier function changes after water cold and hot water exposure and direct contact to cold and hot temperature”. Moreover, we have tried to highlight the new information at the end of each paragraph.
- Do you think repeated exposure of hot or cold water would lead to different conclusions?
Please mention this in the discussion.
Repeated exposure to hot and cold water would lead to higher changes in skin barrier function parameters between baseline and after exposure measurements. Participants were exposed for 10 minutes. If the exposure is longer, it is expected a greater skin barrier disruption compared to baseline. Regarding differences between hot and cold water, we believe that longer exposure would have not changed the difference between them, as skin barrier disfunction would increase with both exposure. This information has been added in the discussion: “… as longer exposure would lead to even greater changes in skin barrier function parameters.”
- You mentioned in the abstract that proper temperatur e for hand washing is important to avoid skin barrier damage. Please stat e yo ur recommendation of the optimal temperature
We have included our recommendations about water temperature and we have also mention that it would be interesting to conduct more studies to select properly the water temperature. The following sentence has been added: “It would be advisable to use cold or lukewarm water for handwashing and avoid hot water. Further research with different water temperature ranges should be conducted to select the most adequate temperature for handwashing”.
- Figure legends (contain brief in formation of the figures) are needed in each figure.
We have included a more explicative title and larger footnotes in each figure:
Figure 3. Skin barrier function parameters on the hands at baseline and after hot and cold water exposure.
AU, arbitrary units; SCH, stratum corneum hydration; TEWL, transepidermal water loss.
*p-value after using Student’s t-test for paired samples to compare skin barrier function between baseline and after cold water exposure. **p-value after using Student’s t-test for paired samples to compare skin barrier function between baseline and after hot water exposure. ***p-value after using Student’s t-test for paired samples to compare skin barrier function after cold and hot water exposure. Only significative p-values are shown (p<0.05)
Figure 4. Skin barrier function parameters on the volar forearm at baseline and after direct contact to different temperatures.
AU, arbitrary units; SCH, stratum corneum hydration; TEWL, transepidermal water loss.
*p-value after using Student’s t-test for paired samples to compare skin barrier function between baseline and after 4ºC exposure. **p-value after using Student’s t-test for paired samples to compare skin barrier function between baseline and after 44ºC exposure. ***p-value after using Student’s t-test for paired samples to compare skin barrier function after 4ºC and 44ºC exposure. Only significative p-values are shown (p<0.05)
- Line 222-223: “water disrupt the skin”; line 204: “Water exposure impairs skin barrier” I think these sentences are not proper. Does any water disrupt/impair skin barrier?
We have changed this sentence to “water exposure damages the skin barrier”
- Abstract: “water immersion impairs skin barrier function” Water immersion has a broad meaning, so I don’t think it is appropriate to mention this due to special condition of your experiments
We have changed this sentence to “water exposure”
Round 2
Reviewer 1 Report
The authors answered my questions.
Author Response
Thank you for the comments
Reviewer 2 Report
No more comments
Author Response
Thank you for the comments
Reviewer 3 Report
Please see attachment.

Author Response
Thank you for all the comments
- Author’s response: Participants were healthcare workers from our hospital, so the results are more homogeneous because different professions are not being compared. A total of 46% (23/50) were doctors, 36% (18/50) were nurses, and 18% (9/50) were miscellaneous healthcare workers, including nurses' assistant and administrative staff. This information has been added in table 1.
My comment: The healthcare workers from same hospital indeed have similar working environment. However, I think doctors and nurses definitely wash hand more frequently than administrative staff. I think it will be beneficial if you can also provide characteristic information of the subjects based on how many times they wash hand in a day.
Thank you for the comment. We did not collect information about the times each participant washed his hand. It should be also considered that we are not comparing two groups of participants, we are comparing participants before and after each exposure, paired samples. So, each participant is assumed to be its own control and the times of hand washing should not have influenced the changes observed. Following your recommendation, we have added the following sentence in our study limitations: It should be also considered that times of hand washing could impact on skin barrier function. This fact is not expected to change our results conclusion because in our study we compared participants before and after each exposure, paired samples, but it should be considered when designing this type of research.
- Table 3, temperature exposure: It seems that the other studies in the table were air exposure with different temperature, which I think is different compare to your study which used direct contact. What do you think about this?
It is true that the exposure type is different, but they were the only research that evaluated the impact of temperature on the skin. We have added this comment in the discussion: “Regarding the direct effect of temperature on the skin, it has previously been observed that ambient air temperature affects skin properties…. We found that temperature, TEWL and erythema increase after exposure to heat. We did not find any differences in pH or SCH, which may be explained by a short contact time or by differences in the type of temperature exposure. Our research assessed the impact on skin barrier function after direct contact with different temperatures while the other evaluate changes after ambient air. We did not find any study that compare epidermal barrier changes after direct contact with different temperatures. It would be interesting to develop further research to compare skin barrier impairment between ambient air exposure and direct contact with temperatures”.
- Line 379: you are comparing your study (with 30 years old as cut off) and other studies(with 65 years old as cut off). I believe it has been known that skin barrier function between 30 and 65years old is different. I don’t think you can compare your study with them.
After reading this comment and comment 4, we decided to omit the information about baseline parameters and focus on changes after water and temperature exposure between sexes and ages. So, this information has been deleted in the new version.
- Author’s response: “As far as we know, no previous study included information regarding changes after water and temperature exposure between sexes and ages. We observed that women were more prone to have changes in temperature and SCH after cold water exposure, being likely explained by the slightly thicker stratum corneum in males [40]”.
My comment: in the first half part of the paragraph (line 377-384) you explained the differences of skin barrier function parameters between sexes and ages (in general). I suggest you put more emphasis on the discussion about the changes after water and temperature exposure between sexes and ages. And I haven’t seen your discussion on the skin barrier function changes after water and temperature exposure based on age.
We have changed these paragraphs and have put more emphasis in the differences after water and temperature exposure between sexes and ages as recommended: “As far as we know, no previous study included information regarding changes after water and temperature exposure between sexes and ages. We observed that women were more prone to have changes in temperature and SCH after cold water exposure, being likely explained by the slightly thicker stratum corneum in males [36]. Anyway, it should be considered that relationship between gender and skin barrier function seems to be outweighed by the individualized factors[36]. In relation to different impact of water and temperature depending on the age, we only found that temperature and pH increases was higher in the younger group after water exposure. This could be explained because young individuals have a greater blood flow, especially in sites exposed to the environment like the hands, what would rapidly adapt the skin to the environmental conditions[37]”.
- Author’s response: “It would be advisable to use cold or lukewarm water for handwashing and avoid hot water. Further research with different water temperature ranges should be conducted to select the most adequate temperature for handwashing”. My comment: I suggest that you put this sentence `It would be advisable to use cold or lukewarm water for handwashing and avoid hot water` in the abstract as well.
This sentence has been added in the abstract, as recommended.
- Author’s response: We have included a more explicative title and larger footnotes in each figure:
My comment: Usually we put more information of the data itself (in short version) instead of the statistic analysis method. Please see other papers fora better description about this.
We have included the following information in the tables: Data is expressed as mean (standard deviations).
- Author’s response: We have changed this sentence to “water exposure”
My comment: I`m still not convenience with the phrase ‘water exposure’ damage skin barrier function. This phrase may give impression that water is not good for skin barrier, while water is necessary to keep our skin function normally and hydrated. You may specifically add information about the water that can disrupt skin barrier.
We have changed this by “long and continuous water exposure could damage the skin”. We are not sure if this is the sentence that you are looking for. Please if you think there is another expression or phrase that would fit better, we would be very grateful if you could provide it to us. Thank you for the comment.
- Please put brackets for figure or table information in the text, e.g. line 99, 110, 358.
Brackets have been added for figures and tables
